# Valorization of biomass into amine-functionalized bio graphene for efficient ciprofloxacin adsorption in water-modeling and optimization study

Seid Kamal Ghadiri[1], Hossein Alidadi[2], Nahid Tavakkoli Nezhad[3], Allahbakhsh Javid[1], Aliakbar Roudbari[1], Seyedeh Solmaz Talebi[4], Ali Akbar Mohammadi[5]*, Mahmoud Shams[2]*, Shahabaldin Rezania[6]*

1 Department of Environmental Health Engineering, School of Public Health, Shahroud University of Medical Sciences, Shahroud, Iran, 2 Social Determinants of Health Research Center, Mashhad University of Medical Sciences, Mashhad, Iran, 3 Department of Environmental Health Engineering, Student Research Committee, School of Health, Mashhad University of Medical Sciences, Mashhad, Iran, 4 Department of Epidemiology, School of Medicine, Shahroud University of Medical Sciences, Shahroud, Iran, 5 Department of Environmental Health Engineering, Neyshabur University of Medical Sciences, Neyshabur, Iran, 6 Department of Environment & Energy, Sejong University, Seoul, South Korea

* mohammadi.eng73@gmail.com (AAM); shamsmh@mums.ac.ir (MS); shahab.rezania@sejong.ac.kr (SR)

**Data Availability Statement:** All relevant data are within the paper and its Supporting Information files.

## Abstract

A green synthesis approach was conducted to prepare amine-functionalized bio-graphene (AFBG) as an efficient and low cost adsorbent that can be obtained from agricultural wastes. In this study, bio-graphene was successfully used to remove Ciprofloxacin (CIP) from synthetic solutions. The efficacy of adsorbent as a function of operating variables (i.e. pH, time, AFBG dose and CIP concentration) was described by a polynomial model. A optimal99.3% experimental removal was achieved by adjusting the mixing time, AFBG dose, pH and CIP concentration to 58.16, 0.99, 7.47, and 52.9, respectively. Kinetic model revealed that CIP diffusion into the internal layers of AFBG controls the rate of the process. Furthermore, the sorption process was in monolayer with a maximum monolayer capacity of 172.6 mg/g. Adsorption also found to be favored under higher CIP concentrations. The thermodynamic parameters ($\Delta G° < 0$, $\Delta H° > 0$, and $\Delta S° > 0$) demonstrated that the process is endothermic and spontaneous in nature. The regeneration study showed that the AFBG could simply regenerated without significant lost in adsorption capacity.

## Introduction

Parallel to the benefits brought, the occurrence of antibiotics and their residues in the environment recognized a serious threat by health care professionals. Global estimates showed an annual 100,000–200,000 tons of antibiotic consumption [1]. Most of them have complex structure that make them resistant to biological decomposition and their accumulative occurrence in the environment [2].Even in trace levels, antibiotics are responsible for emerging the

**Funding:** This work was funded by the Shahroud University of Medical Sciences, Iran (Grant N. 97158).

**Competing interests:** No Conflict of interest.

resistant microbial strains, higher mortality rate, increasing the costs and period of treatment and extending the geographical dimensions of diseases [3].Therefore, antibiotics controls before they discharge into the environment is a high priority practice.

Ciprofloxacin (CIP) is a man-made fluoroquinolone compound that has been detected in several hundred ng/L in water resources across the world [4].The extensive utilization, high solubility and rigidity to biological metabolism, present CIP an important antibiotic that needs further study [5].

Up to now, biodegradation, advance oxidation, adsorption and catalytic degradation, etc., explored by scientists to remove antibiotics [6–8].Although many of these approaches suffer from drawbacks such as low efficiency, high sludge production, longtime and high energy demand, high capital and operational costs. Adsorption is a favorable technique in water and wastewater treatment industry that goes beyond these limitations [9].

For many years, researchers trying to find more sustainable sources of raw material to remove pollutants as adsorbents. Recently, biomass made of corn stove or sugarcane have been used to produce a range of non-food products from biofuels to spandex [10]. Graphene has recently received much attention as an adsorbent due to its high active surface area and various functional groups. Interestingly, graphene can be produced using the thermochemical method of ~~inedible~~ biomass or agricultural waste [11].Corn Stover was used as a feedstock for graphene production due to its high carbon content, inedibility and high worldwide production [12, 13].Amine functionalization of carbon-based materials is common for the removal of various contaminants from aqueous solutions [14–16].

We hypothesized that amine modified graphene (produced from corn stover) could enhance CIP removal from synthetic solutions. To the best of our knowledge, this is the first report on the CIP removal by amine- functionalized bio graphene. A systematic Box-Behnken design (BBD)approach adopted to develop a vigorous statistical model by design of the study by response surface methodology (RSM).

## Materials and methods

### Chemicals and reagents

The used Ciprofloxacin in the experiments was obtained from Sigma-Aldrich. Other reagents and chemicals for the preparation of the adsorbent were purchased from Merck Company.

### Synthesis of graphene from corn stover

For the removal of impurities, first, the obtained corn stover sample swashed thoroughly with deionized water. After drying at 105˚C for 24 h, 50 g of sample was sonicated at 35 kHz for 2 h in a 2L of ethanol solution (50%). After second drying, the sample was combusted at 275˚C for 45 min under the nitrogen flow (5 L/min) to achieve combusted corn stover (CCS).Finally, the combusted corn stover (CCS) was sieved to the size of 80 mesh (<177 μm).

By using thermo-chemical reduction process, CCS was transformed into bio graphene. First, 25 g of CCS was well mixed with a KOH/CCS ratio of 10 for 12 h in 500 mL of de-ionized water. The produced material filtered and washed with de-ionized water to reach to neutral pH. Then after drying and crashing, was activated at 800˚C in an electric furnace under a nitrogen gas (5 L/min) for 2 h.

The amine functionalization of bio graphene was implemented by using grafting amine group method. For preparation of an aminated intermediate solution, a12 mL aliquot of N, N-dimethylformamide (DMF)was reacted with 10 mL of epichlorohydrinat 90˚C for 90 min in a 250 mL three-neck round bottom flask. Then, 5 mL of ethyl ene diamine added to the mixture and stirred at 90˚C for 30 min. Next, 12 mL of trimethylamine added into the mixture and

stirred at 90°C for 120 min. Finally, 10 mL of pyridine and 20 g of bio graphene added to the well-mixed aminated intermediate solution and stirred for 120 min at 75°C.

After filtration, the precipitate was washed with 2 L of ethanol solution (50%), 2 L of NaOH (0.1 M) and 2 L of HCl (0.1 M) and then extensively rinsed with de-ionized water. The product (amine- modified graphene (AFBG)was dried at 70°C for 24 h, sieved to < 177 μm and then used subsequently in the CIP removal experiments. The schematic diagram of synthesis pathway is shown in Fig 1.

## Characterization of the products

Physical and chemical properties of the samples were determined by specific surface area (SSA), X-ray fluorescence (XRF), X-ray diffraction (XRD), scanning electron microscopy (SEM) and transmission electron microscopy (TEM). The specific surface area (SSA) of the samples were measured with a BELSORP-mini-II (BEL Japan, Inc.) at 77°K

## Determination of pHzpc

The pH at zero point of charge (pHzpc) of bio graphene and AFBG was determined using a method as reported by Li et al. [17]. In short, initial pH of a series of batch solutions adjusted to the range of 2–10 (pHi) and 0.1 g of adsorbent added to each flask. After mixing for 24 h, the mixture centrifuged and the solute tested for final pH (pHf). The pHzpc then obtained from plot of ΔpH (pHi—pHf) vs pHi. The pHzpc of bio graphene and AFBG in this study were4.1 and 8.2, respectively.

## Study framework

All the experiments were performed in batch mode in 50 ml flasks containing 25 ml of CIP solution. The CIP removal experimentally was determined under optimum condition to ensure that the model is reliable. Then, kinetic, isotherm and thermodynamic models were set for CIP removal.

Residual CIP concentration in the solutions determined using a High-performance liquid chromatography, (HPLC, Knauersmartline, Germany) equipped with a C8 vortex column and UV detector at 270 nm. The mobile phases were Methanol, HCL 0.01 M and Acetonitrile (12/68/20 v/v). The removal efficiency (μ) was determined from the Eq 1 which described elsewhere [18, 19].

$$\mu\ (\%) = \frac{(C_i - C_f) \times 100}{C_i} \tag{1}$$

Where, Ci is the initial and Cf is the final CIP concentrations (in mg/L), respectively.

Capacity of adsorbent also is a major factor in sorption study, which determined using the Eq 2

$$qe = \frac{(C_i - Ce) \times V}{m} \tag{2}$$

Where, Ce is the CIP concentration at equilibrium, v is the volume of sample (L) and m is the amount of adsorbent added (g), respectively [20].

## Design of experiments

The knowledge about the effect of each individual variable and their interaction on the adsorption efficiency is crucial by efficient design of a sorption unit. To achieve this goal, a systematic

**Fig 1. Schematic diagram of amine- modified graphene (AFBG) synthesis.**

approach must be followed for conducting the experiments. RSM is a useful tool to design the experiments in an organized way to develop a robust mathematical model [21]. The model provides the prediction ability for response when independent parameters changes simultaneously [22, 23]. In Box-Behnken design (BBD), a standard RSM, each independent variable coded according to the following formula:

$$\text{Code value} = \frac{X_A - X_0}{\Delta X}(x) \tag{3}$$

In the Eq 3, XA is the real value of factor, X0 is the real value of factors at the center point (coded as 0) and $\Delta X$ is the step change. In BBD, each variable coded in three levels named center point (level of 0) and factorial points (levels of ±1). Range and levels of CIP concentration (X1), Adsorbent dosage (X2), contact time(X3) and pH(X4) as input factors in this study are presented in Table 1.

Performing the experimental design according to RSM (presented in Table 3) gave a dataset to fit a quadratic polynomial response model (which generally shows as Eq 4) for mathematical prediction of removal efficiency and describing the degree of influence that each independent variable has on response.

$$Y = b_0 + \sum_{i=1}^{k} b_i X_i + \sum_{i=1}^{k} b_{ii} X_i^2 + \sum_{i=1}^{k-1} 1 \sum_{i=2}^{k} b_{ij} X_i X_j + \varepsilon \tag{4}$$

Where, Y is the estimated response (CIP removal), $b_0$ is a constant coefficient, bi, bii, and bij stand for the regression coefficients for linear, quadratic, and interactive parameters, respectively, Xi, and Xj representing the independent terms, and $\varepsilon$ is the random error of the model. The adequacy of the model in predicting the response check by statistical coefficients that discussed in the following.

## Desorption and regeneration experiments

Regeneration study of saturated AFBG was carried out using HCl (0.1 mol/L) solution as eluting agent. For this purpose, 70:1solution/solid ratio of HCl contacted to AFBGfor2 h at 200 rpm. The desorbed CIP then measured in the supernatant. The adsorption/desorption cycle were repeated for four times. The CIP desorption ratio calculated by the below equation:

$$\text{Desorption ratio}(\%) = \frac{\text{amount of CIP ions desorbed}}{\text{amount of CIP ions adsorbed}} \times 100 \tag{5}$$

**Table 1. Range and levels of independent variables used in design of experiments.**

| Factor | | Variable level | | |
|---|---|---|---|---|
| | Code value | -1 | 0 | + 1 |
| CIP concentration (mg/L) | $X_1$ | 10 | 55 | 100 |
| Adsorbentdosage (g/L) | $X_2$ | 0.1 | 0.55 | 1 |
| Contact time (min) | $X_3$ | 5 | 32.5 | 60 |
| pH | $X_4$ | 4 | 8 | 12 |

**Table 3. BBD design of experiments for CIP removal.**

| Run No | Coded variable | | | | Response (% removal) | | Run No | Coded variable | | | | Response (% removal) | |
|---|---|---|---|---|---|---|---|---|---|---|---|---|---|
| | X1 | X2 | X3 | X4 | Observed | Predicted | | X1 | X2 | X3 | X4 | Observed | predicted |
| 1 | -1 | 0 | -1 | 0 | 86.1 | 82.8 | 16 | -1 | 0 | 0 | 1 | 65.3 | 68.7 |
| 2 | 0 | 0 | 0 | 0 | 86.6 | 84.1 | 17 | 0 | -1 | -1 | 0 | 46.6 | 63.8 |
| 3 | 0 | 0 | -1 | 1 | 55.0 | 56.5 | 18 | 0 | 0 | 1 | 1 | 64.0 | 59.0 |
| 4 | 0 | 1 | -1 | 0 | 88.3 | 66.6 | 19 | 0 | -1 | 0 | 1 | 39.8 | 57.1 |
| 5 | -1 | 1 | 0 | 0 | 95.2 | 78.9 | 20 | 0 | 1 | 0 | 1 | 73.5 | 56.0 |
| 6 | 0 | 1 | 1 | 0 | 94.9 | 79.2 | 21 | -1 | 0 | 0 | -1 | 96.3 | 97.5 |
| 7 | 1 | 0 | 0 | -1 | 55.3 | 53.4 | 22 | 0 | 0 | 0 | 0 | 86.0 | 84.1 |
| 8 | 1 | 0 | 0 | 1 | 38.8 | 39.1 | 23 | 0 | 0 | 1 | -1 | 93.4 | 88.4 |
| 9 | 1 | -1 | 0 | 0 | 24.3 | 37.1 | 24 | 0 | -1 | 1 | 0 | 48.9 | 72.1 |
| 10 | 1 | 1 | 0 | 0 | 80.7 | 59.2 | 25 | -1 | 0 | 1 | 0 | 92.6 | 89.7 |
| 11 | 0 | -1 | 0 | -1 | 53.1 | 72.6 | 26 | 0 | 0 | 0 | 0 | 81.4 | 84.1 |
| 12 | 0 | 0 | -1 | -1 | 68.5 | 70.0 | 27 | -1 | -1 | 0 | 0 | 73.2 | 91.2 |
| 13 | 0 | 0 | 0 | 0 | 82.4 | 84.1 | 28 | 0 | 1 | 0 | -1 | 98.8 | 83.5 |
| 14 | 1 | 0 | 1 | 0 | 51.0 | 56.4 | 29 | 1 | 0 | -1 | 0 | 37.4 | 42.3 |
| 15 | 0 | 0 | 0 | 0 | 84.2 | 84.1 | | | | | | | |

## Results and discussion

### Characterization of the products

Based on the Brunauer, Emmettand Teller (BET) equation [24], the obtained SSA of corn stover (<177 μm), combusted corn stover at 275°C (<177 μm) and bio grapheme were 2.82, 37.78, and 493.54 m$^2$/g, respectively.

The chemical composition of adsorbent which determined by XRF is shown in Table 2. The main chemical components of the corn stover were silicon dioxide (34.60%) and alumina (5.84%)

Fig 2. shows the XRD pattern of raw corn stover that has multiple peaks including a short, broad diffraction peak at around 14° and a higher peak at around16°, which attributed to cellulose as the base component of corn stover. Cellulose also make a higher, broad, peak at around 22° and a shorter peak at around 2theta = 34°.According to Fig 2(A), the base material in this study is completely natural and is consistent with corn stover characteristics reported previously [25].

As shown in Fig 2(B), by combustion of corn stover at 270°C, the major peaks associated with the cellulose constituents have been completely altered and replaced with minerals, which are presented as impurities accompanying with the carbon molecules. According to Table 1,

**Table 2. Chemical composition of corn stover (w/w).**

| Component | (w/w%) | Component | (w/w%) |
|---|---|---|---|
| Silicon Dioxide (SiO$_2$) | 34.60 | Sulfur Trioxide (SO$_3$) | 0.57 |
| Alumina (Al$_2$O$_3$) | 5.84 | Titanium Dioxide (TiO$_2$) | 0.42 |
| Potassium Oxide (K$_2$O) | 4.14 | Sudium Oxide (Na$_2$O) | 0.19 |
| Calcium Oxide (CaO) | 3.55 | Manganese Oxide (MnO) | 0.11 |
| Iron (III) Oxide (Fe$_2$O$_3$) | 2.81 | Strontium Oxide (SrO) | 0.03 |
| Magnesium Oxide (MgO) | 2.51 | Barium Oxide (BaO) | 0.01 |
| Phosphorus Pentoxide (P$_2$O$_5$) | 2.28 | Loss on ignition | 42.94 |

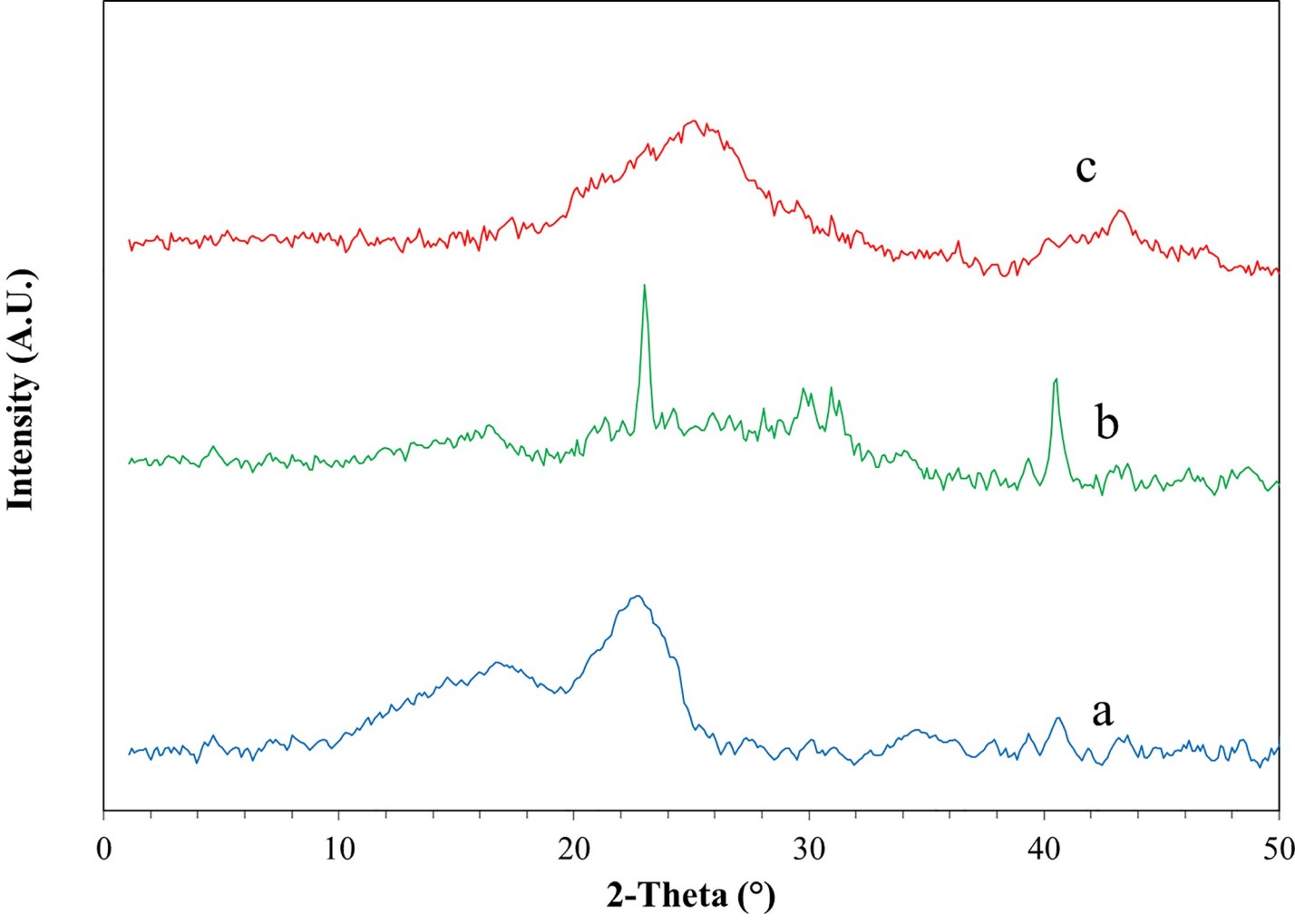

**Fig 2.** The XRD images of (a) raw corn stover (b) combusted corn stover at 275˚C (CCS) and (c) bio graphene.

most of the mineral impurities in the corn stover were silica, potassium, aluminum, and calcium, which in in accordance with corresponding peaks in Fig 2B [26].Fig 2(C) Is the XRD pattern of bio graphene synthesized by the thermo-chemical processing of combusted corn stover. The thermo-chemical process aimed to remove impurities of corn stover and to increases the spacing between carbon plates by penetration of KOH into the interlayer spaces [11, 27].Finally, the heat generated in the process further reduces the carbon plates and showed the XRD pattern of bio graphene with a high and broad diffraction peak centered at 25.5˚. The XRD pattern of the bio graphen was similar to the pattern presented by other studies for reduced graphene oxide (RGO) [28, 29].

Fig 3 shows the SEM image of synthesized bio graphene from corn stover and functionalized bio grapheme using grafting amine groups method. The images were taken at 15 kV intensity. According to the 50 μm image scale, the transparencies of bio graphene nano sheets (a) were clearly visible. The uniform texture of the bio graphene Nano sheets indicated the appropriate final wash and synthesis of the product without the presence of salt crystals and other impurities [30–33].Moreover, smoother surface area of AFBG Nano sheets compared to

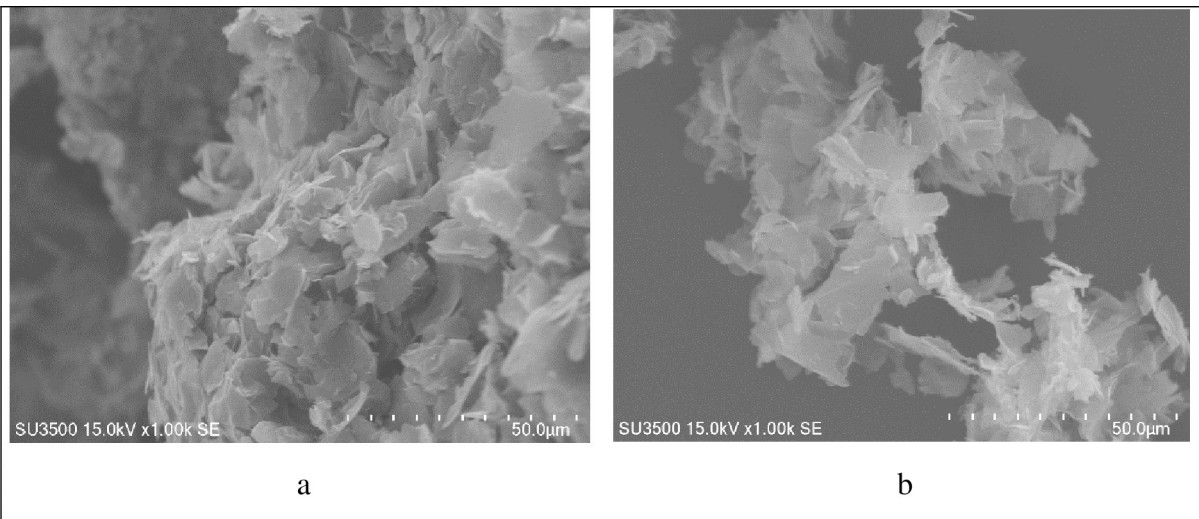

**Fig 3.** SEM image of the (a) bio graphene obtained from corn stover using thermo-chemical process and (b) functionalized bio grapheme using grafting amine group's method.

raw nano bio graphene indicated appropriate modification of the material by grafting amine groups [34].

Fig 4, shows the TEM image of nano-bio-graphene synthesized from the corn stover. The figure show the number of layers formed by the thermo-chemical reduction process of the corn stover. Shadows from the edges of synthesized bio graphene at 1.5 and 2.5 μm scales represent the number of formed layers clearly. As can be seen, the synthesized bio graphene was considered as bilayer or low-layer graphene. In addition, the images show the synthesis of a graphene with smooth and clean edges which is a characteristics of a successful synthesis pathway [27].

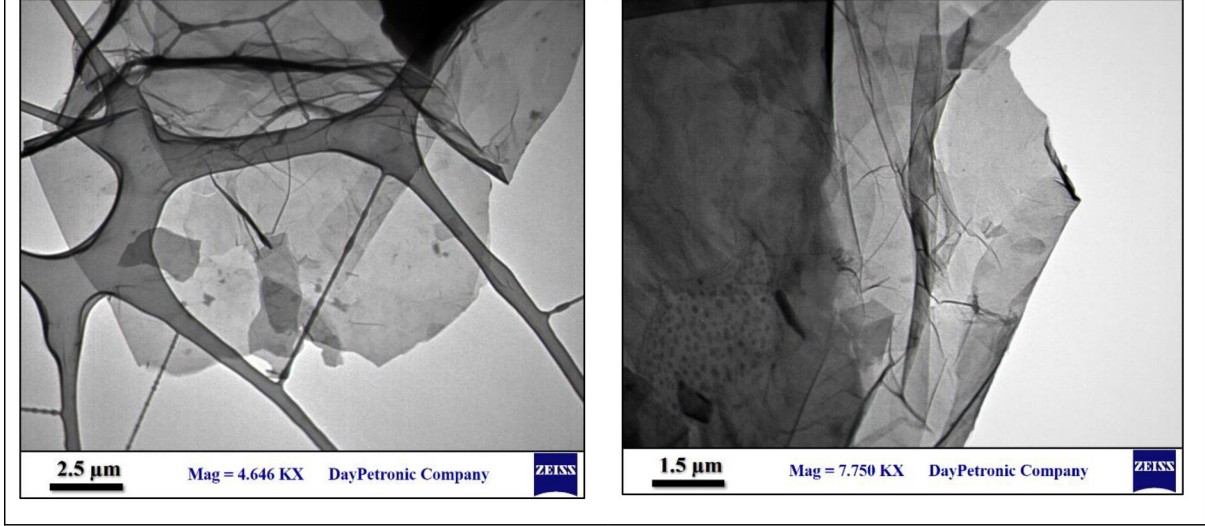

**Fig 4. TEM images with different magnifications of nano-bio graphene synthesized by thermo-chemical reduction method.**

## Process modeling

As presented in Table 3, the study design employed 29 runs which performed in triplicate and the average values were used for statistical analysis and developing a quadratic response model.

Sequential linear, 2Fl, quadratic and cubic models were fitted to the experimental data to compare the suitability of fitted models. According to the F-value and p-value in the Table 4, the quadratic model could describe the data well.

RSM use Fisher's test and Student test (F- test and t-test, respectively) to evaluate the inter-action effects and individual effect of variables on response. In a statistical viewpoint, the significance of a model term directly increased by value of t and conversely related to p value [35].

The changes in response as a function of change in dependent variables explained by ANOVA test. Three statistical parameters namely correlation factor ($R^2$), adjusted correlation factor ($R_{adj}^2$), lack of fit (LOF), were used to evaluate the suitability of model. As shown in the Table 5, both $R^2$ (0.98) and $R_{adj}^2$ (0.95) are close to 1 and within ± 0.2 of each other. LOF value is also non-significant (0.0707), which collectively indicates that the model is statistically adequate.

Based on Table 5, $X_1$, $X_2$, $X_3$, X4, $X_1X_2$, $X_1^2$, $X_2^2$, $X_3^2$ and $X_4^2$ were significant terms of model. To provide more concise and useful model that includes the insignificant terms, all the terms in Table 5 included in the quadratic model for CIP removal. Eq 5 present the quadratic equation obtained by RSM, in term of coded values:

CIP removal (%)

$$
\begin{aligned}
= 84.1 &- 18.43\ X_1 + 2.46\ X_2 + 5.24\ X_3 - 10.75\ X_4 + 8.6X_1X_2 + 1.78X_1X_3 \\
&+ 3.63\ X_1X_4 + 1.08\ X_2X_3 - 3\ X_2X_4 - 3.97\ X_3X_4 - 10.07\ X_1^2 - 7.42\ X_2^2 - 6.25\ X_3^2 \\
&- 9.37\ X_4^2
\end{aligned}
\tag{6}
$$

The direct or indirect effect and the level of influence on CIP removal for each term is simply realizable from the positive or negative sign and the magnitude of the term in the equation. It can simply ascertained from the Fig 5.that the observed points on the plot were distributed relatively next to the regression line, indicating the developed model provides a good estimation for the experimental data.

## Effect of variables and their interactions

The CIP removal efficiency is a function of operational variables and their possible interactions. Fig 6 and 6A–6C visualized the effects of independent variables and their interactions, as showed in the Table 5. Adsorbent dose has the highest coefficient in the Eq 5, which means it was most influential parameter in the removal efficiency. Having the second large coefficient in the equation, initial CIP concentration recognized as the second important variable that control the rate of CIP adsorption. The effect of initial CIP concentration on adsorbent performance was depicted in Fig 6.pH has been literally nominated as an important environmental factor that govern the chemistry of processes. It could influence the ionization nature and charge of adsorbent, contaminant and co-occurrence ions in water. As presented in Fig 6, the highest CIP removal occurred at around neutral and acidic pH. An explanation for this behavior is the ionic state and the interactions between amine groups on AFBG and carboxylic groups on CIP. In particular, under acidic conditions the amino groups protonated ($-NH_3^+$) which in turn hamper their ionic interaction with acid contaminants (i.e., containing a carboxyl group) like CIP [36–38].

**Table 4. Sequential model sum of squares.**

| Model formula | df | Sum Sq. | Mean Sq. | F value | Pr(>F) |
|---|---|---|---|---|---|
| Mean vs Total | 1 | 143714.6 | 143714.6 | | |
| Linear vs Mean | 4 | 10816.4 | 2704.1 | 33.5 | < 0.0001 |
| 2FI vs Linear | 6 | 464.8 | 77.5 | 0.9 | 0.4863 |
| Quadratic vs 2FI | 4 | 1208.1 | 302.0 | 16.1 | < 0.0001 |
| Cubic vs Quadratic | 8 | 206.4 | 25.8 | 2.7 | 0.1155 |
| Residual | 6 | 56.9 | 9.5 | | |

pHzpc of AFBG and molecular structure of CIP could also describe the phenomena behind these findings. pHzpc of AFBG was8.2 which means the charge of AFBG was positive, neutral and negative, at pH<8.2, pH = 8.2 and pH>8.2, respectively. Molecular structure of CIP is presented in Table 6 and showed that protonation–deprotonation reactions in CIP structure makes CIP mainly cationic at pH < 5.9, zwitterion state under 5.9 < pH < 8.9 and anionic at pH > 8.9.Thus, the electrostatic repulsion force between CIP ions and AFBG would be predominant under strong acidic and alkaline condition.

In addition, Fig 6 also shows the effect of contact time on CIP sorption. based on the results, CIP removal was quite fast at the beginning, however, the adsorption rate becomes slow while the number of the active sites on the surface occupied by time.

## Model optimization and confirmation

As mentioned earlier, performing the experiments according to RSM, provides a mathematical model that enable researcher to optimize the process. For the current work, the highest

**Table 5. Analysis of variance (ANOVA) for CIP removal.**

| Source | Sum of Squares | df | Mean Square | F Value | p-value Prob> F |
|---|---|---|---|---|---|
| Model | 12489.4 | 14.0 | 892.1 | 47.4 | < 0.0001 |
| X1 | 4077.5 | 1.0 | 4077.5 | 216.8 | < 0.0001 |
| X2 | 5022.5 | 1.0 | 5022.5 | 267.1 | < 0.0001 |
| X3 | 329.7 | 1.0 | 329.7 | 17.5 | 0.0 |
| X4 | 1386.8 | 1.0 | 1386.8 | 73.8 | < 0.0001 |
| X1.X2 | 295.8 | 1.0 | 295.8 | 15.7 | 0.0 |
| X1.X3 | 12.6 | 1 | 12.6 | 0.67 | 0.4267 |
| X1.X4 | 52.56 | 1 | 52.56 | 2.8 | 0.1167 |
| X2.X3 | 4.62 | 1 | 4.62 | 0.25 | 0.6277 |
| X2.X4 | 36 | 1 | 36 | 1.91 | 0.1881 |
| X2.X4 | 63.2 | 1 | 63.2 | 3.36 | 0.0881 |
| X1^2 | 657.33 | 1 | 657.33 | 34.96 | < 0.0001 |
| X2^2 | 358.01 | 1 | 358.01 | 19.04 | 0.0006 |
| X3^2 | 253.72 | 1 | 253.72 | 13.49 | 0.0025 |
| X4^2 | 569.09 | 1 | 569.09 | 30.26 | < 0.0001 |
| Residual | 263.26 | 14 | 18.8 | | |
| Lack of Fit | 243.22 | 10 | 24.32 | 4.85 | 0.0707 |
| Pure Error | 20.04 | 4 | 5.01 | | |
| Cor Total | 12752.65 | 28 | | | |
| Std. Dev. | 4.34 | R-Squared | | 0.9794 | |
| Mean | 70.4 | Adj R-Squared | | 0.9587 | |
| C.V. % | 6.16 | Pred R-Squared | | 0.8877 | |
| PRESS | 1432.24 | Adeq Precision | | 26.421 | |

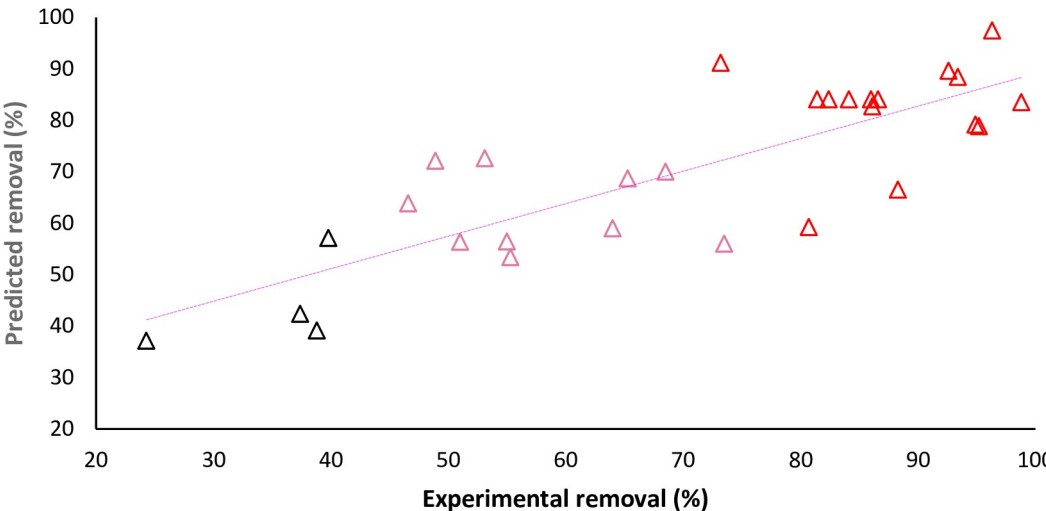

**Fig 5. Experimental vs model predicted efficiency for CIP removal by AFBG.**

adsorption efficiency was considered as the target for model optimization and the range of variables set as they applied in the study (Table 1).Table 7 shows the level of individual variables in which the process could proceed in optimum condition. The removal efficiency under optimum condition predicted to be 99.99%. To evaluate the accuracy of optimization, the parameters in Table 7 were simulated and the CIP removal were measured in triplicate. In addition, the average experimental removal at optimum (99.3) was quite close to that predicted by the model.

## Adsorption isotherm and kinetic modeling

Adsorption isotherm and kinetic modeling are essential part of sorption study that help elucidate important information regarding the adsorption mechanism, adsorbent surface and its affinity to specific contaminant. Isotherm equations model the mobility of adsorbate molecules on the surface of adsorbent under constant environmental condition. On the other hand, kinetic models, applied to find which step in kinetic theory including bulk transportation, film transportation, intra-particle diffusion and adsorption controls the adsorption process. The optimum level of 7.47 and 0.99 g for pH and adsorbent dose were used in the equilibrium and kinetic experiments. Table 8 shows four widely used isotherm and three well-known kinetic models used to fit the data. Modeled kinetic data also illustrated in Fig 7.

The model parameters and coefficients for the isotherms are presented in Table 8.According to the $R^2$ values, the data were simulated by isotherm model in the order of Langmuir, Freundlich, Temkin and Dubinin–Radushkevich [41–43].

Table 8 also shows the correlation coefficients for Intra-particle diffusion model is superior to other models. This result simply confirmed that the dominant and rate-limiting mechanism for CIP adsorption was migration of CIP molecules from liquid bulk to adsorbent pores. Larger Kp value at higher CIP concentration in intraparticle diffusion model indicated the greater sorption inhibition by thickened boundary layer.

Owing to the better conformity of the sorption with the Langmuir isotherm, it can deduced that the CIP covered a monolayer on AFBG. Result also confirmed that the AFBG surface composed of a uniformly distributed sorption site.

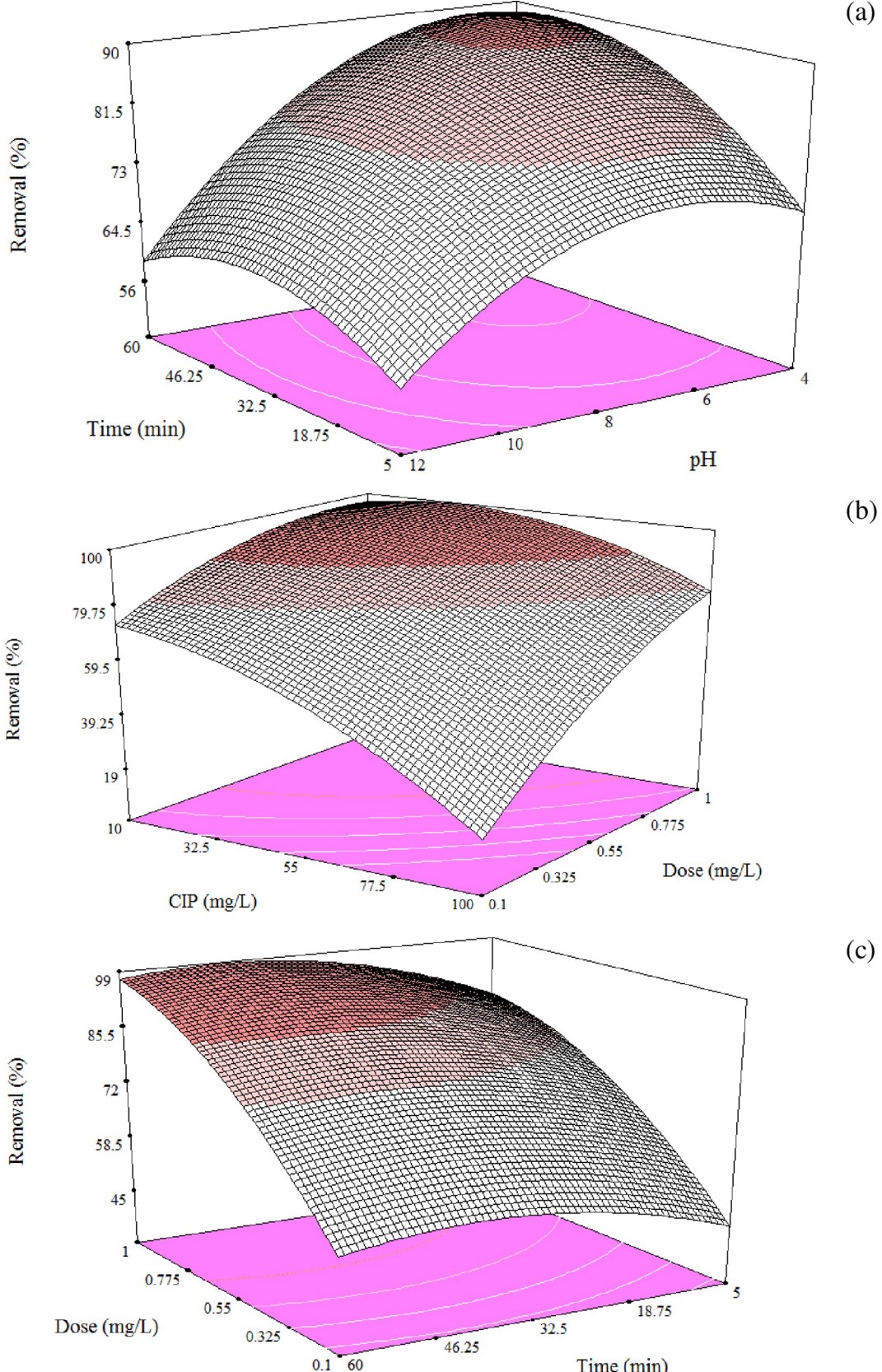

**Fig 6. CIP removal as a function of operating variables, (a) the effect of pH and time (b) CIP concentration and adsorbent dose, and (c) adsorbent dose and agitation time.**

**Table 6. Structural and chemical properties of ciprofloxacin and its pKa [39].**

| Ciprofloxacin structure | Molecular formula | pKa |
|---|---|---|
|  | $C_{17}H_{18}FN_3O_3$ | $pKa_1 = 5.9$ |
| | | $pKa_2 = 8.9$ |

**Table 7. Optimum values for each independent variable obtained from model optimization.**

| Factor | Time(min) | Adsorbent (g/L) | pH | CIP(mg/L) | Removal (%) | |
|---|---|---|---|---|---|---|
| | | | | | Predicted | Experimental |
| Value | 58.16 | 0.99 | 7.47 | 52.9 | 99.99 | 99.3 |

**Table 8. The kinetic and isotherm parameters fitted for CIP adsorption [40].**

| Kinetic Model | Linear Form | Parameter | Value | | |
|---|---|---|---|---|---|
| | | | 50 mg/L | 75 mg/L | 100 mg/L |
| Pseudo- first order | $Log(q_e - q_t) = logq_e - \frac{k_1}{2.303}.t$ | $q_{e,cal}$ [mg/g] | 36.4 | 83.35 | 121.1 |
| | | $K_1$ [min$^{-1}$] | -0.12 | -0.13 | 0.13 |
| | | $R^2$ | 0.97 | 0.92 | 0.91 |
| Pseudo- second order | $\frac{t}{q_t} = \frac{1}{k_2 qe^2} + \frac{1}{qe}.t$ | $q_{e,cal}$ [mg/g] | 54.7 | 82.4 | 115.7 |
| | | $K_2$ [min$^{-1}$] | 0.01 | 0.00 | 0.00 |
| | | $R^2$ | 0.99 | 0.97 | 0.9 |
| Intra-particle diffusion | $q_t = k_p.t^{0.5}+c$ | $K_p$ [mg/g. min$^{-0.5}$] | 4.53 | 11.3 | 15.9 |
| | | $R^2$ | 0.99 | 0.97 | 0.98 |
| **Isotherm model** | **Linear Form** | **Parameter** | **Value** | | |
| Langmuir | $\frac{Ce}{qe} = \frac{1}{qm}Ce + \frac{1}{qmb}$ | $q_{max}$ (mg/g) | 172.6 | | |
| | | $K_L$ (L/mg) | 0.7 | | |
| | | $R^2$ | 0.975 | | |
| Freundlich | $Log\, q_e = log\, K_F + \frac{1}{n}log\, C_e$ | $K_F$ mg/g(L/mg)$^{1/n}$ | 89.07 | | |
| | | N | 5.3 | | |
| | | $R^2$ | 0.971 | | |
| Temkin | $q_e = B_1 ln.k_t + B_1 lnC_e$ | $k_t$ (L/mg) | 0.77 | | |
| | | $B_1$ | 18.19 | | |
| | | $R^2$ | 0.91 | | |
| Dubinin–Radushkevich | $\ln q_e = \ln q_m - \beta \epsilon^2$ | $q_{max}$ (mg/g) | 120.5 | | |
| | | $\beta$ | 1.6 | | |
| | | $R^2$ | 0.66 | | |

Adsorption capacity is a key factor in the economy of an adsorbent utilization [44]. A unique tool for comparing the capacity of adsorbents for a specific contaminant described by Langmuir as $q_{max}$. The $q_{max}$ for AFBG and other adsorbents reported in the literature are shown in the Table 9. AFBG showed a remarkable adsorption capacity compared to many adsorbents reported.

Eq 6 present a dimensionless constant named separation factor (RL), derived from the Langmuir model, that expressed the essential characteristics of the Langmuir as unfavorable

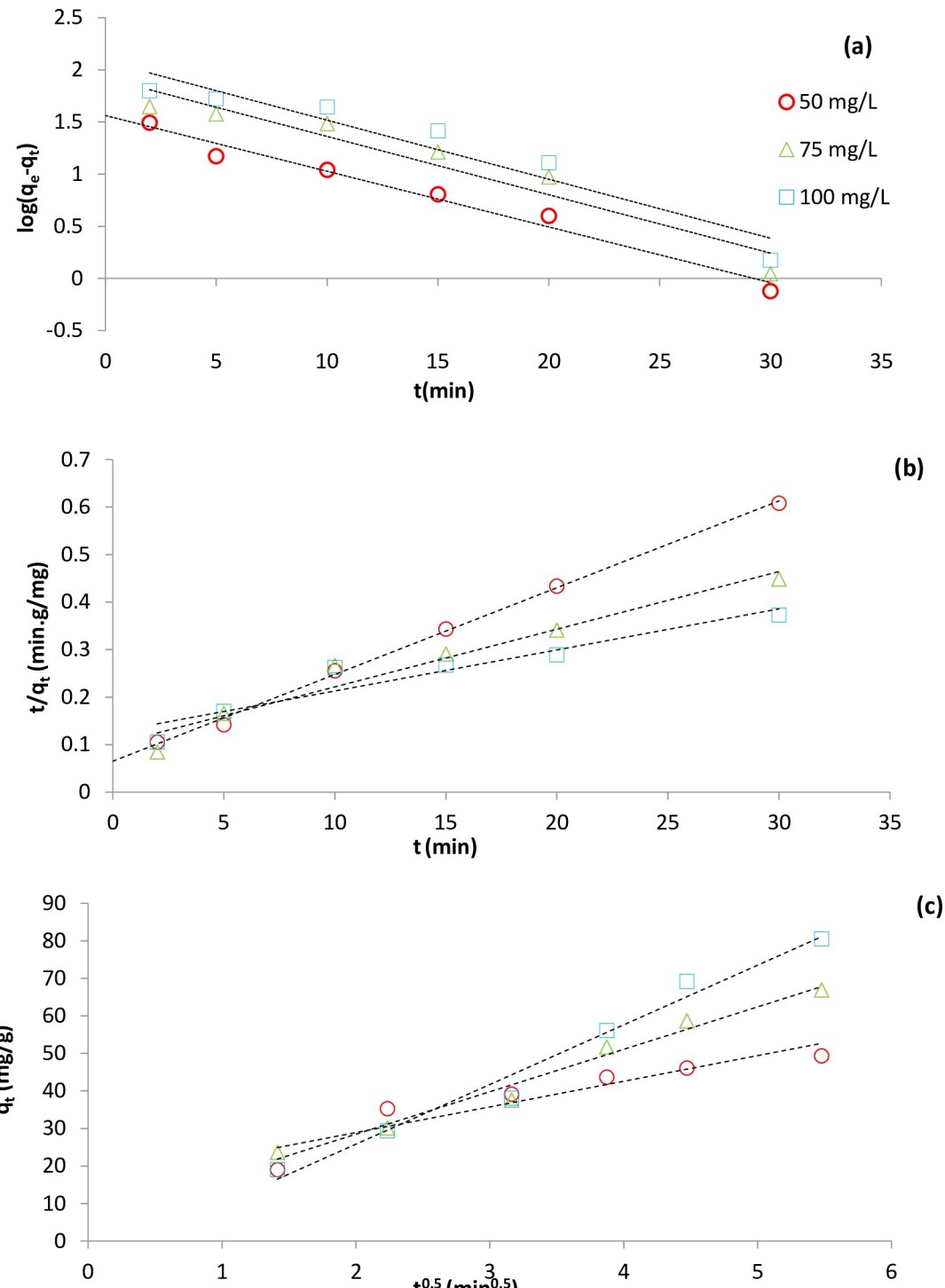

**Fig 7. Linear plots of kinetic models used for adsorption of CIP (a) Pseudo- first order, (b) Pseudo- second order and (c) Intra-particle diffusion.**

**Table 9. Comparison of maximum monolayer capacity of adsorbents for CIP.**

| Adsorbent | $q_{max}$ (mg/g) | Reference | Adsorbent | $q_{max}$ (mg/g) | Reference |
|---|---|---|---|---|---|
| Pretreated oat hulls | 83 | [45] | Fe3o4/go/citrus peel-derived magnetic bio-nanocomposite | 283.4 | [46] |
| KOH modified biochar | 23.36 | [47] | Graphene hydrogel | 348 | [48] |
| Fe-doped MCM-41 | 136.9 | [49] | Nanotube structured hallo site | 21.7 | [50] |
| Carbon prepared from sulphuric acid carbonization date palm leaflets | 133.3 | [51] | Calotropis gigantea fiber | 77.3 | [52] |
| Fe3O4 nanoparticles | 24 | [53] | Cnts/L-cys@GO/SA triple-network composite hydrogel | 200 | [54] |
| Graphene oxide/calcium alginate | 39.06 | [55] | Amine- functionalized bio graphene | 172.6 | Current study |

(RL > 1), linear (RL = 1), favorable (0 < RL < 1), or irreversible (RL = 0)

$$R_L = \frac{1}{1 + K_L C_0} \tag{7}$$

Where, $K_L$ and Co are Langmuir constant (L/mg) and adsorbate concentration (mg/L), respectively. As seen in the Fig 8, there was a slight decrease in RL from 0.02 to 0.007 as the CIP concentration increased from 50 to 200 mg/L, which revealed the adsorption was more favorable at higher concentrations.

## Thermodynamic study

Standard enthalpy (ΔH°), standard entropy (ΔS°) and Gibb's free energy (ΔG°) which are the most important parameters in thermodynamic study were calculated from the following equations

$$\Delta G° = -RTlnK_L \tag{8}$$

$$lnK_L = \frac{\Delta S}{R} - \frac{\Delta H°}{RT} \tag{9}$$

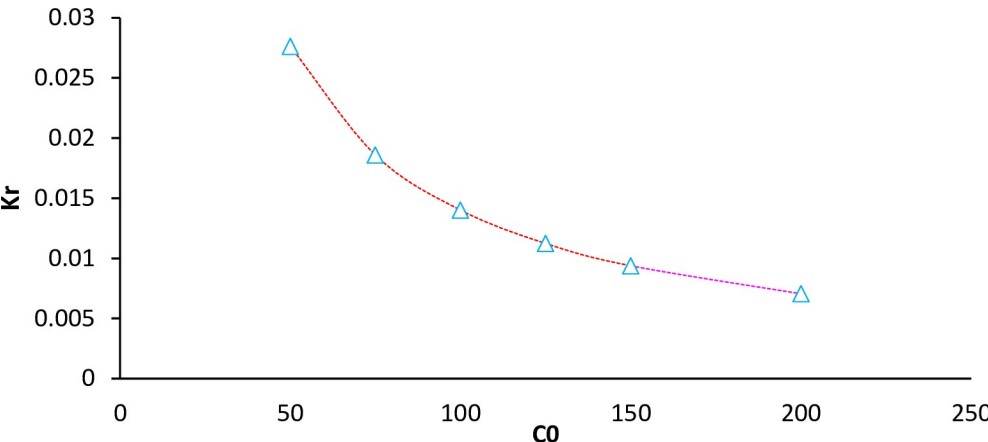

**Fig 8. Plot of Kr versus CIP concentration.**

Table 10. Thermodynamic parameters for CIP adsorption.

| Temperature K | Ce mg/L | -$\Delta G°$ kJ/mol | $\Delta H°$ KJ/mol | $\Delta S°$ KJ/mol.K |
|---|---|---|---|---|
| 293 | 2.15 | -7.5 | 45.8 | 0.18 |
| 303 | 1.7 | -8.6 | | |
| 313 | 0.6 | -11.7 | | |
| 323 | 0.5 | -12.6 | | |

Where, R and T are universal gas constant (8.314 J/mol. K) and temperature (K), respectively. The Thermodynamic parameters could be obtained from the linear plot of ln $k_0$ vs 1/T. As shown in Table 10, the negative values of Gibbs energies ($\Delta G°$) demonstrate that the process going spontaneous. The positive sign of $\Delta H°$ and $\Delta S°$ confirmed the endothermic nature of adsorption.

## Regeneration study

For determination of desorption ability and reusability of AFBG, desorption and regeneration tests were carried out. Table 11 shows the results of four series of CIP adsorption/ desorption cycle by AFBG. As mentioned earlier, HCl 0.1 mol/L solution was used as eluting agent for regeneration purpose. Based on the results, after four sequential adsorption–regeneration cycles, the adsorption capacity of AFBG for CIP just reduced 2.6 mg/g. Therefore, a 0.1 mol HCl /L with a solution/solid ratio of 70:1 was an effective eluent agent for regeneration purposes. The regeneration test also proved that the electrostatic attraction (between the positive charged of AFBG and negative charged ciprofloxacin) was the predominant mechanism for CIP removal.

## Conclusion

In this study, a green synthesis approach was adopted to valorize a low cost and affluent agricultural material for excellent ciprofloxacin removal. Corn stover in a series of thermal processing converted to carbon and then to graphene, where further decorated with amine. The efficacy of amine- functionalized bio graphene (AFBG) for CIP removal modeled as a function of pH, time, AFBG dose and CIP concentration. The model then optimized for maximizing CIP removal by AFBG. To clarify the mechanism of the sorption process, Kinetic, isotherm and thermodynamic modeling analysis performed. The Langmuir isotherm model showed superior adsorption capacity of 172.6 mg CIP /g AFBG. Adsorption also favored at higher CIP concentrations and temperature. The study also shows the AFBG could easily regenerate through a simple acid wash process without considerable lost of adsorption capacity. In conclusion, AFBG shows excellent physical and sorption properties and thus is promising material to be research and applied for pharmaceutical wastewater treatment.

Table 11. Desorption ratio of ciprofloxacin and reusability of AFBG parameters.

| Regeneration order | Adsorption capacity (mg/g) | Desorption ratio (%) |
|---|---|---|
| Primery adsorbent | 172.6 | - |
| 1 | 170.9 | 98.65 |
| 2 | 169.2 | 98.03 |
| 3 | 168.8 | 97.38 |
| 4 | 168.0 | 96.74 |

## Supporting information

**S1 Table.**
(XLSX)

**S2 Table.**
(XLSX)

**S3 Table.**
(XLS)

## Acknowledgments

The authors acknowledge Shahroud University of Medical Sciences for financial support.

## Author Contributions

**Funding acquisition:** Seid Kamal Ghadiri, Allahbakhsh Javid, Aliakbar Roudbari, Seyedeh Solmaz Talebi, Mahmoud Shams.

**Writing – original draft:** Seid Kamal Ghadiri, Aliakbar Roudbari.

**Writing – review & editing:** Hossein Alidadi, Nahid Tavakkoli Nezhad, Allahbakhsh Javid, Seyedeh Solmaz Talebi, Ali Akbar Mohammadi, Mahmoud Shams, Shahabaldin Rezania.

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
