## [Decision Letter · Decision Letter 0]

18 Feb 2020

PONE-D-20-00322

Valorization of biomass into amine- functionalized bio graphene for efficient Ciprofloxacin removal, modeling and optimization study

PLOS ONE

Dear Dr Mohammadi,

Thank you for submitting your manuscript to PLOS ONE. After careful consideration, we feel that it has merit but does not fully meet PLOS ONE’s publication criteria as it currently stands. Therefore, we invite you to submit a revised version of the manuscript that addresses the points raised during the review process.

We would appreciate receiving your revised manuscript by Apr 03 2020 11:59PM. To enhance the reproducibility of your results, we recommend that if applicable you deposit your laboratory protocols in protocols.io, where a protocol can be assigned its own identifier (DOI) such that it can be cited independently in the future. For instructions see: http://journals.plos.org/plosone/s/submission-guidelines#loc-laboratory-protocols

We look forward to receiving your revised manuscript.

Kind regards,

Mohammad Al-Ghouti

Academic Editor

PLOS ONE

Journal Requirements:

1. In your Methods section, please provide additional information on the origin of the corn stover samples. If these were obtained from markets or stores, please provide the geographic coordinates and names of the purchase locations (e.g., stores, markets), if available, as well as any further details about the purchased items (e.g., lot number, source origin, description of appearance) to ensure reproducibility of the analyses.

Reviewers' comments:

Reviewer's Responses to Questions

**Comments to the Author**

1. Is the manuscript technically sound, and do the data support the conclusions?

Reviewer #1: Yes

Reviewer #2: Partly

2. Has the statistical analysis been performed appropriately and rigorously? 

Reviewer #1: Yes

Reviewer #2: N/A

3. Have the authors made all data underlying the findings in their manuscript fully available?

Reviewer #1: Yes

Reviewer #2: Yes

4. Is the manuscript presented in an intelligible fashion and written in standard English?

Reviewer #1: No

Reviewer #2: Yes

5. Review Comments to the Author

Reviewer #1: The manuscript describes the preparation and utilization of an aminated bio-graphene as adsorbent for removal of ciprofloxacin in water. Results indicate a good performance of the material and a favorable adsorption process.

I believe this work is provides enough data to corroborate conclusions of the authors and has a sufficient degree of novelty, therefore it is worthy of publication. However, a major issue affects the manuscript: it does not deepen and explain the scientific phenomena underlying the adsorption process. The authors should clarify that the amino groups, in particular under acidic conditions, can easily bind acid contaminants (i.e., containing a carboxyl group), like ciprofloxacin. The results obtained by RSM should be elucidated under this point of view, citing suitable literature on aminated adsorbents (e.g., DOI: 10.1016/j.ijbiomac.2020.01.171, 10.1016/j.reactfunctpolym.2019.104354, 10.3390/polym11101701).

In addition to this major issue, others require revision as well:

- The title should explicitly mention that the paper deals with adsorption, hence it might be revised as “Valorization of biomass into amine-functionalized bio-graphene for efficient Ciprofloxacin adsorption in water- modeling and optimization study”.

- English should be carefully revised. Just few examples in the abstract:

* the first two sentences might be revised as: ”A green synthesis approach was adopted to prepare amine-functionalized bio-graphene (AFBG) as an efficient and low cost adsorbent that can be obtained from abundant agricultural wastes. In this study, it was successfully employed to remove Ciprofloxacin (CIP) from water.”

* ln. 32, add “, respectively,” after the numbers

* ln 36, replace the sentence between parentheses with “(<delta>Go<0, <delta>Ho>0, and <delta>So>0)”

- ln 57, DOI: 10.1016/j.watres.2018.04.056 might be a suitable citation here.

- ln 78, DOI: 10.3390/polym11101701 might be a suitable citation here.

- ln 91 revise measure unit to “kHz” (i.e., k in lower case).

- ln 113 and ln 127, the authors affirm that the method was previously reported but no reference is provided.

- ln 162-164, characterization methods should be moved in the ‘Materials and methods’ section.

- ln 325 Kelvin measure unit must be written with capital letter

Reviewer #2: The manuscript titled "Valorization of biomass into amine- functionalized bio graphene for efficient Ciprofloxacin removal, modeling and optimization study" represents some good results but the writing should be revised. The following questions should be addressed before acceptance.

1. English usage and grammar should be carefully checked throughout the manuscript.

2. Please add the schematic diagram of synthesis route that it can show reaction step in details.

3. Fig. 1 in Page 8: 'Fig 1. The XRD images of (a) raw corn stover (b) combusted corn stover at 275°C (CCS) and (c) bio graphene. But the expression at the bottom is ' It can be seen in Fig. 1. (b) that by combustion of corn stover at 270°C'. Please confirm the temperature is 275°C or 270°C.

4. Fig. 2 in Page 9, the SEM image of (a) bio graphene obtained from corn stover using thermo-chemical process and (b) functionalized bio grapheme using grafting amine groups method should be compared at the same image scale.

5. In Page 9, it is claimed that structure of nano-bio graphene synthesized by thermo-chemical reduction method was observed by TEM images. In my opinion, the TEM image of functionalized bio grapheme using grafting amine groups method is recommended.

6. The discussion of the regeneration ability of amine-functionalized bio-Graphene for ciprofloxacin removal is strongly recommended.

7. Some relevant literatures may be cited to support the research significance of this wok, such as:

(1) Chemical Engineering Research and Design, 2020, 154, 192-202;

(2) Industrial and Engineering Chemistry Research, 2019, 58, 17075-17087;

(3) Colloids and Surfaces A: Physicochemical and Engineering Aspects, 2020, 588: 124393.

 **********

6. PLOS authors have the option to publish the peer review history of their article (what does this mean?). If published, this will include your full peer review and any attached files.

Reviewer #1: No

Reviewer #2: No

---

## [Author Response · Author response to Decision Letter 0]

7 Mar 2020

Dear editor of PLOS ONE

I appreciate you considering our paper for publication in the journal. Thanks to reviewers for their nice words and their valuable comments on the manuscript, we tried to consider them carefully. Here, we provided comment to comment responses. The revised manuscript with highlighted changes attached. I hope you find the responses convincing and the revised manuscript appropriate for final publication.

Kind regards

Ali Akbar Mohammadi and Mahmoud Shams, Corresponding author

Responses to Reviewer #1:

Comments 1:

A major issue affects the manuscript: it does not deepen and explain the scientific phenomena underlying the adsorption process. The authors should clarify that the amino groups, in particular under acidic conditions, can easily bind acid contaminants (i.e., containing a carboxyl group), like ciprofloxacin. The results obtained by RSM should be elucidated under this point of view, citing suitable literature on aminated adsorbents (e.g., DOI: 10.1016/j.ijbiomac.2020.01.171,10.1016/j.reactfunctpolym.2019.104354, 10.3390/polym11101701).

Response: First, thank you for your warm words on the present work. Thank you very much for your guidance. We read the valuable papers you suggested in the comment. They were interesting works and we used their scientific information to explain the mechanism behind the adsorption system behavior under acidic/alkaline condition. 

Comments 2:

- The title should explicitly mention that the paper deals with adsorption, hence it might be revised as “Valorization of biomass into amine-functionalized bio-graphene for efficient Ciprofloxacin adsorption in water- modeling and optimization study”.

Response:

Thank you, the title revised according to the comment.

Comments 3:

- English should be carefully revised. Just few examples in the abstract:

* the first two sentences might be revised as: ”A green synthesis approach was adopted to prepare amine-functionalized bio-graphene (AFBG) as an efficient and low cost adsorbent that can be obtained from abundant agricultural wastes. In this study, it was successfully employed to remove Ciprofloxacin (CIP) from water.”

Response:

Thank you, we asked help from our colleague who are fluent in English to revise the entire manuscript language. Reviewer's comments also considered in new version. 

Comments 4:

* ln. 32, add “, respectively,” after the numbers

Response: correction done, thanks.

Comments 5:

* ln 36, replace the sentence between parentheses with “(Go<0, Ho>0, and So>0)”

Response: correction done, thanks.

Comments 6:

- ln 57, DOI: 10.1016/j.watres.2018.04.056 might be a suitable citation here.

Response: Thanks. It was interesting work.

Comments 7:

- ln 78, DOI: 10.3390/polym11101701 might be a suitable citation here.

Response: Thanks. It was interesting work.

Comments 8:

- ln 91 revise measure unit to “kHz” (i.e., k in lower case).

Response: correction done, thanks.

Comments 9:

- ln 113 and ln 127, the authors affirm that the method was previously reported but no reference is provided.

Response: thank you, correction done.

Comments 10:

- ln 162-164, characterization methods should be moved in the ‘Materials and methods’ section.

Response: thank you, correction done.

Comments 11:

- ln 325 Kelvin measure unit must be written with capital letter

Response: thank you, correction done.

Responses to Reviewer #2:

Comments 1:

English usage and grammar should be carefully checked throughout the manuscript.

Response:

Thank you for your interest to the work, we asked help from our colleague who are fluent in English to revise the entire manuscript language and the English of the new version has been improved significantly. 

Comments 2:

Please add the schematic diagram of synthesis route that it can show reaction step in details.

Response:

Sure, thank you. Here we present a schematic diagram of synthesis route and we added it to the manuscript. 

Comments 3:

Fig. 1 in Page 8: 'Fig 1. The XRD images of (a) raw corn stover (b) combusted corn stover at 275°C (CCS) and (c) bio graphene. But the expression at the bottom is ' It can be seen in Fig. 1. (b) that by combustion of corn stover at 270°C'. Please confirm the temperature is 275°C or 270°C.

Response:thank you, correction done.

Comments 4:

Fig. 2 in Page 9, the SEM image of (a) bio graphene obtained from corn stover using thermo-chemical process and (b) functionalized bio grapheme using grafting amine groups method should be compared at the same image scale.

Response:thank you,the SEM image of the samples presented at the same scale.

Comments 5:

In Page 9, it is claimed that structure of nano-bio graphene synthesized by thermo-chemical reduction method was observed by TEM images. In my opinion, the TEM image of functionalized bio grapheme using grafting amine groups method is recommended.

Response:

Thank you. That is good suggestion, but unfortunately, due to some financial constrains, we are not able to have further characterization tests. We hypothesized that the TEM images are usually used to illustrate the integrity of graphene layer edges and the total number of graphene layers formed during the synthesis. In the case of aminated graphene structures , based on the literature, TEM image became dark and the formed layers wouldn’t be detectable.

Comments 6:

The discussion of the regeneration ability of amine-functionalized bio-Graphene for ciprofloxacin removal is strongly recommended.

Response:

Thank you, we conducted regeneration study of the ABGN and the results were given in the manuscript.

Comments 7:

Some relevant literatures may be cited to support the research significance of this wok, such as:

(1) Chemical Engineering Research and Design, 2020, 154, 192-202;

(2) Industrial and Engineering Chemistry Research, 2019, 58, 17075-17087;

(3) Colloids and Surfaces A: Physicochemical and Engineering Aspects, 2020, 588: 124393.

Response:

Thank you, interesting papers. We read them thoroughly and used them to deepen the scientific discussions of the manuscript.

---

## [Decision Letter · Decision Letter 1]

16 Mar 2020

Valorization of biomass into amine- functionalized bio graphene for efficient Ciprofloxacin adsorptionin water-modeling and optimization study

PONE-D-20-00322R1

Dear Dr. Mohammadi,

We are pleased to inform you that your manuscript has been judged scientifically suitable for publication and will be formally accepted for publication once it complies with all outstanding technical requirements.

With kind regards,

Mohammad Al-Ghouti

Academic Editor

PLOS ONE

Additional Editor Comments (optional):

Reviewers' comments:

Reviewer's Responses to Questions

**Comments to the Author**

1. If the authors have adequately addressed your comments raised in a previous round of review and you feel that this manuscript is now acceptable for publication, you may indicate that here to bypass the “Comments to the Author” section, enter your conflict of interest statement in the “Confidential to Editor” section, and submit your "Accept" recommendation.

Reviewer #1: All comments have been addressed

Reviewer #2: All comments have been addressed

2. Is the manuscript technically sound, and do the data support the conclusions?

Reviewer #1: Yes

Reviewer #2: Yes

3. Has the statistical analysis been performed appropriately and rigorously? 

Reviewer #1: Yes

Reviewer #2: Yes

4. Have the authors made all data underlying the findings in their manuscript fully available?

Reviewer #1: Yes

Reviewer #2: Yes

5. Is the manuscript presented in an intelligible fashion and written in standard English?

Reviewer #1: Yes

Reviewer #2: Yes

6. Review Comments to the Author

Reviewer #1: The authors revised all the issues I raised in my previous comments, therefore I suggest acceptance of the manuscript in its present form.

Reviewer #2: (No Response)

7. PLOS authors have the option to publish the peer review history of their article (what does this mean?). If published, this will include your full peer review and any attached files.

Reviewer #1: No

Reviewer #2: No

---

## [Editor Report · Acceptance letter]

23 Mar 2020

PONE-D-20-00322R1 

Valorization of biomass into amine- functionalized bio graphene for efficient Ciprofloxacin adsorption in water-modeling and optimization study 

Dear Dr. Mohammadi:

I am pleased to inform you that your manuscript has been deemed suitable for publication in PLOS ONE. Congratulations! Your manuscript is now with our production department. 

With kind regards,

on behalf of

Dr. Mohammad Al-Ghouti 

Academic Editor

PLOS ONE